An unusual Cretaceous beetle with affinity to Anamorphidae (Coleoptera: Coccinelloidea)

http://orcid.org/0000-0002-9439-202X Li Yan-Da 1 2
Tomaszewska Wioletta 3
Arriaga-Varela Emmanuel 4
Huang Di-Ying 1
Cai Chen-Yang 1 cycai@nigpas.ac.cn
1 State Key Laboratory of Palaeobiology and Stratigraphy, Nanjing Institute of Geology and Palaeontology, Chinese Academy of Sciences , Nanjing , China
2 Bristol Palaeobiology Group, School of Earth Sciences, University of Bristol , Bristol , United Kingdom
3 Museum and Institute of Zoology, Polish Academy of Sciences , Warszawa , Poland
4 Red de Biodiversidad y Sistemática, Instituto de Ecología, A.C. , Xalapa , Mexico
Silva Daniel
Electronic publication date: 2024 Sep 16
Publication date: 2024
Volume: 12
Electronic Location ID: e18008
Received 2024 May 7; Accepted 2024 Aug 9
Copyright: © 2024 Li et al.
Copyright year: 2024
Copyright holder: Li et al.
License: This is an open access article distributed under the terms of the Creative Commons Attribution License, which permits unrestricted use, distribution, reproduction and adaptation in any medium and for any purpose provided that it is properly attributed. For attribution, the original author(s), title, publication source (PeerJ) and either DOI or URL of the article must be cited.
License URL: https://creativecommons.org/licenses/by/4.0/

Keywords: Anamorphidae, Fossil, Phylogeny, Kachin amber, Cretaceous

Funding: National Natural Science Foundation of China 42222201, 42288201 Second Tibetan Plateau Scientific Expedition and Research Project 2019QZKK0706 China Scholarship Council 202108320010 Financial support was provided by the National Natural Science Foundation of China (42222201, 42288201) and the Second Tibetan Plateau Scientific Expedition and Research project (2019QZKK0706). Yan-Da Li is supported by a scholarship granted by the China Scholarship Council (202108320010). The funders had no role in study design, data collection and analysis, decision to publish, or preparation of the manuscript.

==============================
Coccinelloid beetles have a sparse fossil record in the Mesozoic. Here, we describe and illustrate an unusual coccinelloid beetle, Yassibum yoshitomii gen. et sp. nov., from mid-Cretaceous Kachin amber. Yassibum stands out within the Coccinelloidea due to its notched profemora and the presence of antennal grooves on the elytral epipleura. Based on our phylogenetic analyses, we suggest that Yassibum is most likely related to the family Anamorphidae. The alternative placements are critically evaluated based on our comparison of the morphology.

Introduction

Coccinelloidea is a moderately large superfamily in Cucujiformia, with about 10,000 species in 660 genera (Robertson et al., 2015; McKenna et al., 2019). Although previously included in Cucujoidea, coccinelloids have long been recognized as a morphologically derived group called “Cerylonid Series” (Crowson, 1955; Lawrence & Newton, 1982; Ślipiński & Pakaluk, 1991). This group has also been supported by molecular evidence (e.g., Robertson, Whiting & McHugh, 2008; Bocak et al., 2014), and was formally elevated as a superfamily by Robertson et al. (2015).

Although many coccinelloid lineages are likely to have originated during the Jurassic (McKenna et al., 2019; Cai et al., 2022), the Mesozoic fossil record of coccinelloids is quite sparse. The earliest fossils of Coccinelloidea known to date, Archelatrius Kirejtshuk & Azar, Tetrameropsis Kirejtshuk & Azar and Atetrameropsis Kirejtshuk, all originally placed in Latridiidae, were discovered from Early Cretaceous Lebanese amber (Kirejtshuk & Azar, 2008, 2013; Kirejtshuk et al., 2009). Shockley & Alekseev (2014) suggested that the group containing Tetrameropsis and Atetrameropsis might be related to either Latridiidae or Akalyptoischiidae. Recently several coccinelloid fossils have been reported from mid-Cretaceous Kachin amber (Ross, 2024), including representatives of Teredidae (Li, Huang & Cai, 2022), Bothrideridae (Li & Cai, 2024), Corylophidae (Li et al., 2022a), Anamorphidae (Arriaga-Varela et al., 2024) and Endomychidae (Tomaszewska et al., 2018, Tomaszewska, Szawaryn & Arriaga-Varela, 2022; Li, Huang & Cai, 2023; Li et al., 2022b; Arriaga-Varela et al., 2023a). Zippel et al. (2023) also reported possible coccinelloid larvae (which may belong to Endomychidae) from Kachin amber. Other coccinelloid families have yet to be reported from the Mesozoic, but may be found in some Cenozoic deposits, especially Eocene Baltic amber (e.g., Reike et al., 2017; Szawaryn, 2019, 2021; Szawaryn & Kupryjanowicz, 2019; Szawaryn & Tomaszewska, 2020a, 2020b; Bukejs, Kirejtshuk & Rücker, 2011; Bukejs, Reike & Rücker, 2012; Bukejs, Alekseev & Kairišs, 2022; Bukejs et al., 2021; McHugh et al., 2023).

The coccinelloid family Anamorphidae was separated from Endomychidae by Robertson et al. (2015). The family includes over 175 extant species in about 36 genera (Shockley, Tomaszewska & McHugh, 2009a; Shockley, 2010; Narukawa, 2013, 2015, 2017). Anamorphids have been suggested to feed obligately on the spores of fungi, at least in the larval stages (Shockley, Tomaszewska & McHugh, 2009b; Arriaga-Varela, Leschen & Tomaszewska, 2023). Fossils of Anamorphidae were previously known from Baltic amber (Alekseev & Tomaszewska, 2018). Recently Arriaga-Varela et al. (2024) reported the earliest anamorphid fossils from Kachin amber. In this study we describe an unusual coccinelloid fossil from Kachin amber, which could potentially belong to Anamorphidae based on our comparison of morphological characters and formal phylogenetic analyses.

Materials and Methods

Materials

The Kachin (Burmese) amber specimen studied herein (Figs. 1–5) originated from amber mines near Noije Bum (26°20′N, 96°36′E), Hukawng Valley, Kachin State, northern Myanmar, and is deposited in the Nanjing Institute of Geology and Palaeontology (NIGP), Chinese Academy of Sciences, Nanjing, China. The amber piece was trimmed with a small table saw, ground with emery papers of different grit sizes, and finally polished with polishing powder. Jewellery-grade Kachin amber specimens are commonly carried and sold legally in Ruili, Dehong Prefecture on the border between China and Myanmar. The specimen in this study was purchased in 2016 (prior to 2017), and was therefore not involved in the armed conflict and ethnic strife in Myanmar.

Figure 1 General habitus of Yassibum yoshitomii gen. et sp. nov., holotype, NIGP200730, under incident light.

(A) Dorsal view. (B) Ventral view. Scale bars: 500 μm.

Figure 2 General habitus of Yassibum yoshitomii gen. et sp. nov., holotype, NIGP200730, under widefield fluorescence.

(A) Dorsal view. (B) Ventral view. Scale bars: 500 μm.

Figure 3 General habitus of Yassibum yoshitomii gen. et sp. nov., holotype, NIGP200730, under confocal microscopy.

(A) Dorsal view. (B) Ventral view. Scale bars: 500 μm.

Figure 4 Details of Yassibum yoshitomii gen. et sp. nov., holotype, NIGP200730, under confocal microscopy.

(A) Anterior body, ventral view, with arrow indicating the projection on elytral epipleuron. (B) Head, dorsal view. (C) Prothorax, dorsal view, with arrow indicating the basolateral sulcus. (D) Meso- and metathorax, ventral view. Abbreviations: a1–2, antennomeres 1–2; cl, clypeus; ep, elytral epipleuron; lb, labrum; lbp, labial palp; msc, mesocoxa; mtv, metaventrite; mxp, maxillary palp; pc, procoxae; pf, profemur; pn, pronotum. Scale bars: 100 μm.

Figure 5 X-ray microtomographic reconstruction of Yassibum yoshitomii gen. et sp. nov., holotype, NIGP200730.

Left antenna and legs partly removed in (D–F). (A) Dorsal view. (B, F) Ventral view. (C) Lateral view. (D) Anterior view. (E) Posterior view. Scale bar: 500 μm.

Fossil imaging

Photographs under incident light were taken with a Zeiss Discovery V20 stereo microscope. Widefield fluorescence images were captured with a Zeiss Axio Imager.Z2 light microscope combined with a fluorescence imaging system. Confocal images were obtained with a Zeiss LSM710 confocal laser scanning microscope, using the 561 nm (DPSS 561-10) laser excitation line (Fu et al., 2021). Images under incident light and widefield fluorescence were stacked in Helicon Focus 7.0.2 and Zerene Stacker 1.04. Confocal images were stacked with Helicon Focus 7.0.2 and Adobe Photoshop CC. Microtomographic data were obtained with a Zeiss Xradia 520 Versa 3D X-ray microscope at the micro-CT laboratory of NIGP and analyzed in VGStudio MAX 3.0. Scanning parameters were as follows: isotropic voxel size, 1.6339 μm; power, 3 W; acceleration voltage, 40 kV; exposure time, 4 s; projections, 3001. Images were further processed in Adobe Photoshop CC to adjust brightness and contrast.

Phylogenetic analyses

To evaluate the systematic placement of the new fossil, we conducted constrained morphology-based phylogenetic analyses under parsimony (e.g., Li et al., 2023a, 2023b). We applied two matrices to evaluate the phylogenetic position of the new fossil, one based on Lawrence et al. (2011) (Data S1), and the other based on Robertson (2010) (Data S2). The matrix by Lawrence et al. (2011) had broad sampling among the whole Coleoptera, but in the present analysis we only selected the coccinelloid genera and a few cucujiform outgroups. The matrix by Robertson (2010) was focused on Coccinelloidea with higher density of taxon sampling, but had fewer morphological characters. For the new fossil, we successfully coded 134 characters with non-inapplicable states in the matrix based on Lawrence et al. (2011), and 44 characters in the matrix based on Robertson (2010).

During the constrained analyses, the relationships among extant taxa were fixed as the backbone tree according to previous molecular studies, and only the fossil was allowed to move freely. The use of molecular-based constraints would allow a more realistic estimation of the states at ancestral nodes, and therefore contribute to a more authentic placement of the fossil (Fikáček et al., 2020). For the construction of backbone trees, the interfamilial relationships were mainly based on McKenna et al. (2019) and Cai et al. (2022), the positions of Akalyptoischiidae, Mycetaeidae and Eupsilobiidae were based on Arriaga-Varela et al. (2023b), and the intrafamilial relationships not covered by the above studies were mainly based on Robertson et al. (2015) and Che et al. (2021). The position of Loeblioryloninae (Loebliorylon Ślipiński) has not been studied with molecular data and is largely unclear; therefore it was not included in the backbone tree of our analysis.

The analyses were performed under implied weights, using R 4.1.0 (R Core Team, 2021) and the R package TreeSearch 1.3.1 (Smith, 2023). The concavity constant was set to 12, following the suggestion by Goloboff, Torres & Arias (2018) and Smith (2019). In order to inspect both the optimal and suboptimal placements of the fossil, the parsimony scores of the trees with alternative placements of the fossil were mapped to the corresponding branches of the backbone tree (Li et al., 2024a, 2024b). The results were visualized with the R package ggtree 6.5.2 (Yu et al., 2017; Yu, 2020) and graphically edited with Adobe Illustrator CC 2017.

Nomenclature

The electronic version of this article in Portable Document Format (PDF) will represent a published work according to the International Commission on Zoological Nomenclature (ICZN), and hence the new names contained in the electronic version are effectively published under that Code from the electronic edition alone. This published work and the nomenclatural acts it contains have been registered in ZooBank, the online registration system for the ICZN. The ZooBank LSIDs (Life Science Identifiers) can be resolved and the associated information viewed through any standard web browser by appending the LSID to the prefix http://zoobank.org/. The LSID for this publication is: 7F20F1D5-8E2C-407A-A10D-7D169A66EB66. The online version of this work is archived and available from the following digital repositories: PeerJ, PubMed Central SCIE and CLOCKSS.

Systematic paleontology

Order Coleoptera Linnaeus, 1758

Suborder Polyphaga Emery, 1886

Superfamily Coccinelloidea Latreille, 1807

Family (?) Anamorphidae Strohecker, 1953

Genus Yassibum gen. nov.

Type species. Yassibum yoshitomii sp. nov.

Etymology. The generic name is an anagram of Asymbius Gorham, a genus in Anamorphidae. The name is neuter in gender.

Diagnosis. Body oval. Antennae 10-segmented, with 3-segmented club. Labrum with apex deeply emarginate. Pronotal disc with narrow and deep basolateral sulci. Procoxae distinctly projecting. Elytral epipleura wide at base, basally with antennal groove extending posterolaterally. Mesocoxal cavities broadly closed laterally. Mesoventrite and metaventrite without pits near their anterior margins. Metanepisterna concealed. Profemora with prominent incision on outer side. Tarsi simple, 4-4-4.

Yassibum yoshitomii sp. nov.

(Figs. 1–5)

Material. Holotype, NIGP200730.

Etymology. The species is named after the coleopterist Dr. Hiroyuki Yoshitomi.

Locality and horizon. Amber mine located near Noije Bum Village, Tanai Township, Myitkyina District, Kachin State, Myanmar; unnamed horizon, mid-Cretaceous, Upper Albian to Lower Cenomanian.

Diagnosis. As for the genus.

Description. Body oval, with dorsal side weakly convex, about 1.3 mm long, 0.9 mm wide, moderately setose.

Head appearing hypognathous due to angle of attachment between head base and prothorax. Eyes well developed, protuberant, coarsely facetted, without interfacetal setae. Antennae composed of 10 antennomeres; antennomere 1 elongate and relatively wide; antennomere 2 elongate, narrower than antennomere 1, attached to antennomere 1 subapically; antennomere 3 elongate, narrower than antennomere 2; antennomeres 4–7 each less elongate than antennomere 3, about as wide as antennomere 3; antennomere 8–10 elongate and wide, forming distinct club. Antennal sockets possibly visible from above. Antennal grooves not distinctly developed. Frontoclypeal suture likely present, straight. Labrum transverse, distinctly emarginate. Maxillary palps 4-segmented; palpomere 4 flattened and slightly widened apically, with apex obliquely truncate. Labial palps with apical palpomere flattened and enlarged.

Pronotum transverse, laterally slightly explanate; sides gradually converging anteriorly in anterior half; lateral pronotal carinae complete, weakly denticulate; posterior margin broadly and weakly produced backwards medially; anterior angles not produced; disc with narrow raised margins, with a pair of curved narrow basolateral sulci. Prosternal process short, reaching only middle of procoxae. Procoxae longer than wide, strongly projecting.

Scutellar shield small, transverse, anteriorly abruptly elevated. Elytra completely covering abdomen, widest near anterior 1/4, gradually narrowing posteriorly, laterally widely explanate; sides rounded; disc irregularly punctate, with long parasutural stria extending from elytral apex to near scutellum; epipleura wide at base, incomplete towards apices, each basally with well-developed posterolaterally extending antennal groove and anteriorly directed projection near base of antennal groove. Mesoventrite longitudinally carinate at middle, posteriorly bifid, without visible pits or pores. Mesocoxal cavities separated by almost diameter of cavity, laterally broadly closed by meeting of meso- and metaventrites. Metaventrite large, with raised margin anteriorly, with distinct depression at each anterolateral corner (likely for accommodating mesotrochanter), without visible pits or pores near anterior margin. Metanepisterna concealed.

Legs elongate. Trochanterofemoral joint oblique. Profemora with outer edge deeply notched at middle; base of this notch with spine. Tibiae without apical spurs. Tarsi 4-4-4; tarsomeres simple, all much longer than width. Pretarsal claws simple.

Abdomen with six ventrites. Ventrite 1 as long as ventrites 2–4 combined, without postcoxal lines; intercoxal process broadly truncate. Ventrites 2–4 subequal in length. Ventrite 6 apically rounded.

Discussion

Yassibum gen. nov. can be doubtlessly assigned to Coccinelloidea based on its oval and convex body, reduced tarsomeres (4-4-4), and widely separated metacoxae with truncate intercoxal process (Robertson et al., 2015). The strongly reduced hindwing venation is also accordant with the coccinelloid placement (Lawrence et al., 2022). As the character combination of Yassibum does not seem to perfectly fit with any of the existing coccinelloid families, we conducted constrained parsimony analyses under implied weights to test its familial assignment. In the analysis based on the matrix developed by Lawrence et al. (2011), Yassibum is best placed in Anamorphidae, whereas Alexiidae and Endomychidae appear to be the suboptimal alternative placements. If not considering the possibilities of being inserted to the interfamilial branches, the analysis based on the matrix by Robertson (2010) produced overall similar results, with Anamorphidae, Latridiidae, Alexiidae and Endomychidae being the most parsimonious placements for Yassibum.

Alexiidae consists of a single genus, Sphaerosoma Stephens (Sasaji, 1987; Ślipiński & Tomaszewska, 2010; Courtin, Perez & Bouget, 2017; Hinson & Gompel, 2023), which was once included in Endomychidae (Strohecker, 1953; Crowson, 1955). Sphaerosoma shares with Yassibum the 10-segmented antennae, 4-4-4 tarsi and short-oval body shape. However, Sphaerosoma differs from Yassibum in many characters like the pronotum without basal or lateral impressions and with posterior margin almost straight medially, apex of mesoventrite truncate, mesocoxal cavities laterally open, postcoxal lines well-developed on metaventrite and abdominal ventrite 1, hind wings essentially absent, much shorter and stouter tarsi, especially metatarsi, and abdomen with only five ventrites.

Endomychidae is a morphologically diverse family (Tomaszewska, 2000a, 2005, 2010). Many characters of Yassibum can be found in this family. However, some of the characters present in Yassibum seem atypical for Endomychidae sensu stricto (Robertson et al., 2015), including the 10-segmented antennae and laterally broadly closed mesocoxal cavities. Endomychids usually have 11-segmented antennae (except for Merophysiinae, Pleganophorinae and some Endomychinae). Laterally broadly closed mesocoxal cavities were previously thought to be absent in Endomychidae (narrowly closed in Merophysiinae and Pleganophorinae, and open in other subfamilies), and are one of the most important characters separating Anamorphidae from Endomychidae (Tomaszewska, 2000a; Robertson et al., 2015). Nevertheless, the Cretaceous fossil Rhomeocalpsua Li et al. assigned to Endomychidae has been described recently as having broadly closed mesocoxal cavities (Li et al., 2022b), and the extant genus Anamycetaea Strohecker with broadly closed mesocoxal cavities has been recently transferred from Anamorphidae to Endomychidae (Tomaszewska, Szawaryn & Arriaga-Varela, 2023).

The concealed metanepisternum is an unusual character for Yassibum, which is known in Discolomatidae and some Latridiidae (Lawrence et al., 2011). In the analysis based on the matrix by Robertson (2010), the topology also receives a low parsimony score when Yassibum is inserted to the ancestral branch of Latridiidae. However, Yassibum is unlikely to belong to either Discolomatidae or Latridiidae. Discolomatidae differ from Yassibum in the antennae with one-segmented club, presence of gland openings along lateral pronotal and elytral margins, externally small and rounded metacoxae, and abdomen with five ventrites (John, 1959; Cline & Ślipiński, 2010; Cline & Shockley, 2012; Yoshitomi & Pham, 2021). Latridiidae differs from Yassibum in the elongate body shape, 3-3-3 tarsi and often regularly punctuate elytra (Hartley & McHugh, 2010; Lord et al., 2010; Chan & Lee, 2016).

Yassibum appears to fit best in Anamorphidae. The most important character advocating this assignment would be the laterally broadly closed mesocoxal cavities (Sasaji, 1978, 1990; Tomaszewska, 2000a). Yassibum also shares with Anamorphidae (or part of Anamorphidae) the 10-segmented antennae with 3-segmented club (e.g., Parasymbius Arrow, Bystodes Strohecker, Mychothenus Strohecker, Idiophyes Blackburn, Cyrtomychus Kolbe), similar shape of labial palp, pronotum with basolateral sulci, simple 4-4-4 tarsi (e.g., Rhymbomicrus Casey, Micropsephus Gorham, Bystus Guérin-Méneville, Catapotia Thomson, Anagaricophilus Arrow, Symbiotes Redtenbacher), and abdomen with six ventrites (Strohecker, 1953; Tomaszewska, 2000a). Based on the above morphological discussion, we suggest that Yassibum could be placed in the family Anamorphidae, which is also the most parsimonious possibility resolved by our formal analyses (Figs. 6, 7). However, the possibility that Yassibum could represent an extinct family of Coccinelloidea cannot be definitely rejected, as indicated by the analysis (Fig. 7) and the peculiar morphological characters.

Figure 6 Constrained parsimony analysis under implied weights, based on the matrix by Lawrence et al. (2011), showing alternative placements of Yassibum gen. nov.

The score above each branch represents the parsimony score of the topology in which Yassibum is inserted to that branch. The colors are an intuitive display of the parsimony scores.

Figure 7 Constrained parsimony analysis under implied weights, based on the matrix by Robertson (2010), showing alternative placements of Yassibum gen. nov.

The score above each branch represents the parsimony score of the topology in which Yassibum is inserted to that branch. The colors are an intuitive display of the parsimony scores.

The dorsally notched profemur is notable for Yassibum. While in many members of Anamorphidae or Endomychidae the fore, mid or hind legs are notched or toothed, these modifications occur exclusively on the ventral face of the trochanter, femur or tibia (e.g., Tomaszewska, 2000a, 2000b). The procoxae are prominent (associated with the vestigial procoxal process) in some Endomychidae (e.g., Archipines Strohecker or Hylaperdina Tomaszewska; Arriaga-Varela & Tomaszewska, 2016: fig. 10). However, they are not so strongly projecting like in Yassibum. The position and possible use of the modification on the elytral epipleuron is another peculiar feature for Yassibum. At the base of elytral epipleuron, there is a posterolaterally extending groove, and near the base of this groove there is an anteriorly directed projection. Based on the position and shape of this groove, we suppose that it is likely to be used to accommodate the antenna, rather than the profemur. Antennal grooves are commonly found on the head or prothorax of beetles, however, as far as we know, this is the first time for the antennal groove to be discovered on the elytral epipleuron in the whole Coleoptera. A somewhat similar groove on elytral epipleuron is known in the extinct archostematan genus Notocupes Ponomarenko, but in that case it is used for accommodating the mid leg (Li et al., 2023c, 2023d).

Yassibum clearly differs from the other anamorphid beetles known from the mid-Cretaceous, Palaeosymbius spp. (Arriaga-Varela et al., 2024), by the emarginate labrum, profemora with an incision in the exterior margin, mesoventrite without visible pits or pores and with posterior margin of intercoxal process bifid, elytral epipleura excavated, possibly for the reception of the antenna, and metaventrite with anterior margin without pores or pits. While the characters of Palaeosymbius Arriaga-Varela et al. show a very likely affinity with extant genera Symbiotes and Asymbius, the possible position of Yassibum within the anamorphid lineages is yet to be fully understood. Although the overall shape of the pronotum could suggest a resemblance to genera like Bystus and Bystodes, the peculiar characters of the genus described here point out the possibility of a lineage within Anamorphidae that left no descendants in present time.

Supplemental Information

Supplemental Information 1 Character matrix and R script for the phylogenetic analysis based on Lawrence et al. (2011).

Supplemental Information 2 Character matrix and R script for the phylogenetic analysis based on Robertson (2010).

We are grateful to Su-Ping Wu for help with micro-CT reconstruction, Rong Huang for help with confocal microscopy, and Jing-Jing Tang for help with widefield microscopy. Three reviewers provided helpful comments on the manuscript.

Additional Information and Declarations

Competing Interests

Author Contributions

Data Availability

New Species Registration

The authors declare that they have no competing interests.

Yan-Da Li conceived and designed the experiments, performed the experiments, analyzed the data, prepared figures and/or tables, authored or reviewed drafts of the article, and approved the final draft.

Wioletta Tomaszewska analyzed the data, authored or reviewed drafts of the article, and approved the final draft.

Emmanuel Arriaga-Varela analyzed the data, authored or reviewed drafts of the article, and approved the final draft.

Di-Ying Huang analyzed the data, authored or reviewed drafts of the article, and approved the final draft.

Chen-Yang Cai conceived and designed the experiments, analyzed the data, authored or reviewed drafts of the article, and approved the final draft.

The following information was supplied regarding data availability:

The data for the phylogenetic analyses are available in the Supplemental Files.

The original confocal and micro-CT data are available at Zenodo: Li, Y.-D., Tomaszewska, W., Arriaga-Varela, E., Huang, D., & Cai, C. (2024). Confocal and micro-CT data of Yassibum yoshitomii, holotype, NIGP200730 [Data set]. Zenodo. https://doi.org/10.5281/zenodo.11122582.

The following information was supplied regarding the registration of a newly described species:

Publication LSID: urn:lsid:zoobank.org:pub:7F20F1D5-8E2C-407A-A10D-7D169A66EB66.

Yassibum: urn:lsid:zoobank.org:act:EEC2BC57-0E54-4103-85C7-1ABE6361A43A

Yassibum yoshitomii: urn:lsid:zoobank.org:act:255BF78E-4DA1-4859-A449-E705446C670D.

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
