# Peer review of "An unusual Cretaceous beetle with affinity to Anamorphidae (Coleoptera: Coccinelloidea)"

_PeerJ, doi:10.7717/peerj.18008_

## Round 0.1 · original submission · Major Revisions

Dear Dr. Li,

After this first review round, all three reviewers believe your manuscript deserves to be published in PeerJ after a thorough review. Please pay attention to all the issues raised by the reviewers.

Sincerely,
Daniel Silva

Reviewer 1 ·

Basic reporting

It is an unusual of coccinelloid beetle Yassibum yoshitomii gen. et sp. nov., suggesting match to the family Anamorphidae, from mid-Cretaceous Kachin amber. Detailed descriptions and photographs from stereo microscope and CT scanning of X-ray microscope are given, with showing the peculiar morphological characters. The paper is well written and the discussion is appropriate.

Experimental design

no comment

Validity of the findings

no comment

Reviewer 2 ·

Basic reporting

The manuscript's language is clear and understandable. The literature is cited according to the recent advancements in the field. The structure is standard and the figures are appropriate. The only minor issue here is that the description should be in telegraphic style, e.g. without articles. They are present in some lines (e.g. 145, 167). Also, add gen. nov. after each mention of Yassibum.

Experimental design

I have one major concern here: running the maximum parsimony with both constrained clades and implied weighting seems to be too limited. Was there an attempt to run it without constraining and with equal weighting? If the matrix does not contain characters bearing phylogenetic signal at the family level, how does the fossil position was supposed to be reliably resolved?

Validity of the findings

The study is an important addition to our understanding of the evolution of Coccinelloidea and represents a novel finding. I very much enjoyed the Discussion where different possibilities regarding the placement of the fossil were discussed. This was actually even more convincing than the carried analyses because I found them too limited (see my previous comment). After improving that part, I think the manuscript should definitely be published.

Additional comments

It is a nice piece of work and I would be glad to see it published after minor revision.

Reviewer 3 ·

Basic reporting

No issues with English.
Regarding completeness of literature cited I have an impression that certain publications are intentionally omitted. In recent years many papers dealing with dating of divergence time of beetles were published, majority of them contain Coccinelloidea taxa and propose various results. However, these publications are not cited here, and in my opinion they should be (e.g. Zhang et al. 2018; McKenna et al. 2015; Toussaint et al. 2017).
In general structure of the article is correct, however, there is no section "Results" (Systematic paleontology is just a part of results) where authors should describe in detail results from their phylogenetic analyses.

Experimental design

I noticed several issues with the text. Some of them are here briefly discussed, some additional minor are mentioned directly in the attached pdf.
a) Regarding phylogenetic analyses, I am not certain why authors used results from McKenna et al 2015 and Cai et al. 2022 as a backbone topology for their analyses. These large-scale analyses were not focused on Coccinelloidea relationships and contain just a few coccinelloid taxa (McKenna - 5, Cai - 32, and non monophyletic Cerylonidae) with very merely resolved topology in that area. While paper dedicated for Coccinelloidea Robertson's et al. 2015 with global sampling, with majority of Coccinelloidea higher taxa represented by several specimens is ignored. This is not explained.
Moreover, it is not clearly written how many characters were coded for the fossil taxon in each matrix (two sets analyzed). While for Robertson matrix over 50% of characters were coded, for the matrix published by Lawrence et al (over 500 characters coded) only around 25% were used. Therefore, result form the second analysis is not reliable. I would suggest to re-calculate that matrix after deletion of characters with missing data for the fossil taxon. That may give better results, may not, but this is not discussed. Nonetheless, estimation of phylogenetic placement of the fossil taxon with 75% of missing data is irrelevant. Critical selection of morphological characters is essential for this kind of analysis.

b) Regarding nomenclatural act, since PeerJ is just an online journal a direct ZooBank code for each new name must be published along with the description to be compliant with the ICZN regulations.

c) Description of the new taxon must be reviewed. The diagnosis for the new genus is unacceptable as it is not fulfilling ICZN code. Set of characters used in diagnosis, mostly plesiomorphies, do not allow to distinguish newly described taxon from numerous other similar-looking coccinelloid taxa. While most characteristic features are completely omitted.

d) Although, authors attached numerous illustration figs 1-3 are showing almost the same, with just little difference. On the other hand, most important character for the newly described fossil are very scarcely illustrated or not illustrated at all. I suggest o add closeups of the such features as: elytral antennal grove, profemoral notch, tarsus, prosternum and prosternal process.

Validity of the findings

The newly described taxon is certainly interesting finding worth publication after revision.

Additional comments

minor additional comments in pdf

Annotated reviews are not available for download in order to protect the identity of reviewers who chose to remain anonymous.

---

## Round 0.2 · Minor Revisions

Dear Dr. Li,

The raw CT data are mentioned to be available through Zenodo in the reviewer materials but this is not mentioned in the manuscript. This information should be explicitly mentioned in the manuscript and available in final form at the latest upon publication. Please can you make the addition to the text and confirm that it will be available, and then your article can be Accepted

Sincerely,

Daniel Silva

Reviewer 1 ·

Basic reporting

The paper has been good written.

Experimental design

The materials and methods are credibly. The authors have done a very good job for investigating and describing the material.

Validity of the findings

Coccinelloid beetles have a sparse fossil record in the Mesozoic. This paper contributes valuable datas with the morphology features and phylogenetic analyses on systematic of Coccinelloidea.

Additional comments

no comments

Reviewer 2 ·

Basic reporting

The authors accommodated the required changes and/or explained their reasoning. I am satisfied with the current version and recommend it for publication.

Experimental design

All is well explained.

Validity of the findings

The finding is valid and important.

Additional comments

Congratulations to the authors.

Reviewer 3 ·

Basic reporting

.

Experimental design

.

Validity of the findings

.

Additional comments

.

---

## Round 0.3 · accepted · Accept

Dear Dr. Li,

I am pleased to Accept your manuscript for publication in PeerJ.

Sincerely,
Daniel Silva